# The Prognostic Value of PI-RADS Score in CyberKnife Ultra-Hypofractionated Radiotherapy for Localized Prostate Cancer

**DOI:** 10.3390/cancers14071613

**Published:** 2022-03-23

**Authors:** Marcin Miszczyk, Justyna Rembak-Szynkiewicz, Łukasz Magrowski, Konrad Stawiski, Agnieszka Namysł-Kaletka, Aleksandra Napieralska, Małgorzata Kraszkiewicz, Grzegorz Woźniak, Małgorzata Stąpór-Fudzińska, Grzegorz Głowacki, Benjamin Pradere, Ekaterina Laukhtina, Paweł Rajwa, Wojciech Majewski

**Affiliations:** 1IIIrd Radiotherapy and Chemotherapy Department, Maria Skłodowska-Curie National Research Institute of Oncology, 44-102 Gliwice, Poland; lukasz.magrowski@gmail.com; 2Radiology Department, Maria Skłodowska-Curie National Research Institute of Oncology, 44-102 Gliwice, Poland; justyna.rembak-szynkiewicz@io.gliwice.pl; 3Department of Biostatistics and Translational Medicine, Medical University of Łódź, 90-419 Łódź, Poland; konrad@konsta.com.pl; 4Radiotherapy Department, Maria Skłodowska-Curie National Research Institute of Oncology, 44-102 Gliwice, Poland; agnieszka.namysl-kaletka@io.gliwice.pl (A.N.-K.); aleksandra.napieralska@io.gliwice.pl (A.N.); malgorzata.kraszkiewicz@io.gliwice.pl (M.K.); grzegorz.wozniak@io.gliwice.pl (G.W.); malgorzata.stapor-fudzinska@io.gliwice.pl (M.S.-F.); sekretariat.tomaszow@nu-med.pl (G.G.); benjaminpradere@gmail.com (B.P.); wojciech.majewski@io.gliwice.pl (W.M.); 5Department of Urology, Medical University of Vienna, 1090 Vienna, Austria; katyalaukhtina@gmail.com (E.L.); pawelgrajwa@gmail.com (P.R.); 6Institute for Urology and Reproductive Health, Sechenov University, 119435 Moscow, Russia; 7Department of Urology, Medical University of Silesia, 41-800 Zabrze, Poland

**Keywords:** prostate cancer, Pi-Rads, radiotherapy, prognosis, metastasis-free survival

## Abstract

**Simple Summary:**

Magnetic resonance imaging is commonly used in pre-treatment prostate cancer diagnostics. The assessment includes a five-stage scale classification called Prostate Imaging-Reporting and Data System (PI-RADS), routinely used to describe the probability of finding a clinically significant cancer. Less is known about the association of PI-RADS score with patients’ prognosis. Our retrospective study aimed to assess the association between pre-treatment PI-RADS score and risk of developing metastases, based on a cohort of 152 patients treated with ultra-hypofractionated CyberKnife radiotherapy for low or intermediate-risk group prostate cancer. PI-RADS score and the size of the target lesion proved to be significantly associated with the risk of developing metastases, suggesting that the introduction of PI-RADS score to initial risk assessment could improve the patient-tailored management of prostate cancer.

**Abstract:**

Prostate Imaging-Reporting and Data System (PI-RADS) has been widely implemented as a diagnostic tool for significant prostate cancer (PCa); less is known about its prognostic value, especially in the setting of primary radiotherapy. We aimed to analyze the association between PI-RADS v. 2.1 classification and risk of metastases, based on a group of 152 patients treated with ultra-hypofractionated stereotactic CyberKnife radiotherapy for localized low or intermediate risk-group prostate cancer. We found that all distant failures (n = 5) occurred in patients diagnosed with a PI-RADS score of 5, and axial measurements of the target lesion were associated with the risk of developing metastases (*p* < 0.001). The best risk stratification model (based on a combination of greatest dimension, the product of multiplication of PI-RADS target lesion axial measurements, and age) achieved a c-index of 0.903 (bootstrap-validated bias-corrected 95% CI: 0.848–0.901). This creates a hypothesis that PI-RADS 5 and the size of the target lesion are important prognostic factors in early-stage PCa patients and should be considered as an adverse prognostic measure for patients undergoing early treatment such as radiation or focal therapy.

## 1. Introduction

The technical advances in medical imaging and improvements in its reading led to the widespread implementation of magnetic resonance imaging (MRI) as means to improve PCa detection and reduce unnecessary prostate biopsies. Furthermore, the MRI has been incorporated in the modern guidelines for routine target volume delineation in external beam radiotherapy [1], resulting in significantly lower inter-observer variability, smaller target volumes [2,3], and potential reduction of treatment toxicity [4].

The introduction of multiparametric MRI, comprising of anatomical T2-weighted (T2W), functional dynamic contrast-enhanced (DCE), and diffusion weight imaging (DWI) sequence led to significant improvement in sensitivity and specificity of predicting clinically significant PCa. The five-tier diagnostic tool—Prostate Imaging-Reporting and Data System (PI-RADS) was developed to unify the interpretation of prostate MRI. Currently, the updated version (PIRADS v2.1) [5] is the most utilized diagnostic tool for prostate MRI assessment, due to broad validation, good reproducibility, and high diagnostic estimates [6,7,8]. Furthermore, recent evidence suggests PI-RADS may improve PCa staging [9,10] and local treatment planning [5].

While there is overwhelming evidence of PI-RADS diagnostic value, there is a knowledge gap on its association with oncologic outcomes, especially in the context of radiation therapy. Only one study reported its prognostic value in patients treated with different schedules of primary RT [11]. In radiation therapy, PI-RADS could help to improve pre-treatment patient selection for tailored definitive therapy. Thus, we aimed to analyze the prognostic value of pre-treatment prostate MRI PI-RADS v2.1 based on a cohort of patients treated with ultra-hypofractionated CyberKnife radiotherapy for early-stage PCa [12].

## 2. Materials and Methods

### 2.1. Study Group

We performed a retrospective analysis of an institutional registry of 500 consecutive patients treated with ultra-hypofractionated stereotactic CyberKnife RT for localized N0 NCCN low- or intermediate risk-group PCa between 2011 and 2017 [12]. After the exclusion of 271 patients due to initiation of androgen deprivation therapy (ADT) before MRI, and 77 patients due to missing MRI series, the final analysis included 152 patients.

### 2.2. Pi-Rads Score

The PI-RADS score was assessed according to the PI-RADS version 2.1 [13] by the second author (JRS), a radiologist with approximately 20 years of experience, specializing in PCa imaging and diagnostics. The radiologist was blinded to the risk factors and oncological outcomes of the patients. The classifications were made anew based on the pre-treatment prostate MRI performed on 3T scanner in approximately one out of three of the cases (Siemens MAGNETOM Prisma and Philips Achieva), and 1.5T in the remaining two out of three (Siemens MAGNETOM Aera). Any uncertainties were resolved through a consensus agreement with another radiologist. The report included data on the two-dimensional measurements (largest and perpendicular measurement on transverse T2W series); the number of the index lesions, their localization and laterality within the prostate; assessment of T2, DWI, and DCE series; restriction of contrast enhancement; and final PI-RADS score.

### 2.3. Treatment Protocol and Follow-Up

The patients were treated according to the local treatment protocol of primary CyberKnife radiosurgery for localized low or intermediate-risk group PCa according to D’Amico risk classification [14], maximum primary Gleason score of 3 + 4, and two-dimensional prostate diameter below or equal to 5 cm. Each patient received 36.25 Gy in five fractions delivered every other day on a CyberKnife linear accelerator. Three Gold Anchor fiducials were implanted to the prostate before radiotherapy planning CT, and after MRI, as a target for real-time tracking during each treatment session. The patients were positioned on a vacuum mattress, with a moderately full bladder and empty rectum. The target volume and organs at risk were defined on the treatment planning CT, with the aid of a fused treatment planning MRI. The clinical target volume (CTV) included the prostate gland and proximal 1 cm of seminal vesicles. The planning target volume (PTV) was expanded by 3 mm posteriorly and by 5 mm in the remaining directions. None of the patients received pelvic lymph node irradiation. The 6 months ADT was prescribed in intermediate-risk patients at the attending physician’s discretion, depending on the patient’s preferences and clinical risk factors. The follow-up details were extracted from patients’ medical records and the data from the National Cancer Registry. The follow-up visits were scheduled at 1, 4, and 8 months, and every 6 months thereafter, according to the institutional protocol.

The metastases-free survival (MFS) was defined as the time from the initiation of radiotherapy to the occurrence of any metastasis or PCa-related death. The remaining patients were censored with the date of the last known follow-up.

### 2.4. Statistical Analysis

Survival analysis utilized the Kaplan–Meier estimator. Median follow-up was calculated using a reverse Kaplan–Meier estimate.

Univariable analysis was performed with Cox proportional-hazards modeling. Hazard ratios along with their 95% CI were assessed. However, due to a low number of events per variable [15], multivariable analysis was omitted. Instead, to assess the predictive power of a combination of features, we developed a complex modeling pipeline utilizing conditional inference trees [16]. First, due to the low number of events, we balanced the dataset with the synthetic minority oversampling technique (SMOTE), similarly to Ishaq et al. [17]. In the next step, we inducted a conditional inference tree on the whole dataset. To minimize the risk of overfitting, the modeling was restricted to the maximum depth of the tree equal to 2 (i.e., to a maximum of 4 new recursive partitioning classes). Stop criterion was based on multiplicity-adjusted Monte-Carlo *p*-values computed following a “min-p” approach [18].

Although this approach inherently employed embedded feature selection, we performed best subset selection involving all combinations of two and three predictors. For each subset, a conditional inference tree was developed with the same modeling characteristics as described above. The model was scored on the whole original dataset and the Cox proportional-hazards model was developed with predicted recursive partitioning classes as predictors. Performance of the model was assessed based on Harrell’s c-index [19] with a maximal value of 1.0 indicating perfect concordance. Internal validation of models was performed with 100 bootstrap samples to provide bias-corrected indexes. Ninety-five percent CI of bootstrap-based internal validation was calculated based on 100 repetitions of this random process. The best model was selected based on the greatest internal validation c-index. The analysis was conducted in R programming language with appropriate packages available through CRAN. To support the idea of reproducible research, the source code for feature selection and model development with internal validation was published on Github [20].

## 3. Results

The median age was 67 years (IQR 62–73), and the median follow-up (FU) time was 53.6 months (IQR 28.3–70.8 months) in our study group of 152 patients treated with CyberKnife radiosurgery for early-stage PCa. PI-RADS scores of ≤3, 4, and 5 were found in 22 (14.5%), 57 (37.5%), and 73 (48%) of the patients, respectively. Metastases occurred in five patients throughout FU and were primarily located in pelvic lymph nodes (3) or bones (2). In two cases, they were diagnosed through Prostate-Specific Membrane Antigen Positron Emission Tomography (PSMA-PET), in two other cases by 18-Fluor Choline PET, and in one case through bone scintigraphy. In each case, diagnostic imagining was performed due to rising PSA concentration. At the time of data collection, 17 patients (11.2%) were dead. The patients’ clinical characteristics are presented in detail in Table 1.

All the metastases (n = 5/152) occurred in patients with a PI-RADS score of 5 as shown in Figure 1. However, likely due to a low number of events, the log-rank test did not show the significant differences in MFS. Metastases-free median survival time was not reached in follow-up.

Univariable analysis showed that the greatest (HR 1.15; 95% CI 1.06–1.25; per 1 mm increase) and the perpendicular (HR 1.16; 95% CI 1.05–1.127; per 1 mm increase) axial measurement of the index lesion were significantly associated with MFS. Unsurprisingly, the surface area of the index lesion (defined as the product of multiplication of axial measurements) was a single significant factor shortening MFS (HR 1.0029; 95% CI: 1.0011–1.0046, *p* = 0.0013; per 1 mm^2^ increase). 

Firstly, utilizing all the possible predictors with embedded feature selection, we developed a metastases-free survival risk stratification model (conditional inference tree; Figure 2A). The product of axial measurements was the most valuable predictor and allowed for risk stratification characterized by a c-index of 0.871 and 95% CI of internally validated bias-corrected c-index between 0.811 and 0.877.

The best subset evaluation was presented in Table 2. The best model (Figure 2B) was based on a combination of greatest dimension, the product of multiplication of PI-RADS target lesion, axial measurements, and age, achieving a c-index of 0.903 and 95% CI of internally validated bias-corrected c-index between 0.848 and 0.901.

## 4. Discussion

The significance of the PI-RADS score as a prognostic factor is an emerging topic in the PCa literature. However, up to now there has been relatively sparse data analyzing the association between PI-RADS and oncologic outcomes after primary definitive therapy [21]. The available data have been mostly focused on the risk of biochemical failure, recently found to be an obsolete intermediate endpoint in PCa trials [22].

The pivotal finding in our study was that all of the distant failures occurred in patients with PI-RADS scores of 5, and high axial measurements of the target lesion. In fact, in patients with a surface area of target lesion of >352 mm^2^, simply defined as the multiplication of two axial measurements, there was almost 40% risk of developing metastases. Of note, since all of the distant failures occurred in patients presenting a PI-RADS score of 5, the area of the target lesions is probably of less importance in PCa patients with PI-RADS score ≤ 4.

The implementation of PI-RADS to available prognostic models could allow for improved risk stratification and cost-effectiveness [23]. In radiation therapy, the MRI diagnostic scheme may allow discerning low and intermediate risk-group PCa patients with increased risk of developing distant metastases. Similarly, PI-RADS could be used as another means of assuring the patient that active surveillance is the treatment of choice, in patients with neglectable risk of distant failure. The score most likely goes beyond the standard risk assessment through significant association with routinely non-reported histopathological hallmarks [24] and genomic changes [25,26,27,28]. Moreover, the PI-RADS score of 5 and large axial measurements of the target lesion found in low or intermediate-risk group PCa patients could be an important predictive factor for the necessity of secondary pre-treatment biopsy or early confirmatory biopsy in patients opting for active surveillance.

Similar findings were reported earlier concerning the prognostic value of the largest axial measurement of the index lesion [11,29] in patients treated with radiotherapy. The authors showed that the largest axial measurement, along with extraprostatic extension and seminal vesical invasion, are significantly associated with freedom from biochemical failure [29]. The second analysis showed that both index lesion size (>15 mm) and PI-RADS score are also adversely associated with freedom from distant metastases. Applied to our database, the cut-off of 15 mm would allow distinguishing patients into a group with estimated 5-year freedom from distant metastases of 98%, compared to 92.5% for patients with index lesion >15 mm (*p* = 0.22). Importantly, Turchan et al. focused on advanced PCa, in a significantly non-homogenous study group, including RT doses and delivery methods, pelvic lymph node irradiation, ADT, and so on. Our study provides data supporting the prognostic value of Pi-Rads in patients with early-stage localized PCa, and with minimal treatment-related confounding variables, confirming the earlier findings based on a homogenous study group.

The largest axial measurement [9] and PI-RADS score [30,31,32,33] were shown to be correlated with outcomes in patients undergoing radical prostatectomy as well. Some authors suggested interesting derivative indices, such as tumor volume [34], tumor visibility on MR [35], relative lesion volume (calculated as the ratio of lesion volume to prostate volume on MRI) [9], zonal location of the index lesion [36], or MR-based nomogram [31]. The choice between (single or double) axial measurements or volume assessment most likely comes down to the compromise between precision, reproducibility, and time investment, as the tumor volume should likely be regarded as the reference index [37].

Of note, the PI-RADS score was not created for prognostic, but rather a diagnostic purpose, and the available scale was not developed to have optimal cut-offs for best prognostic value. For example, while T2-weighted imaging reflects higher cellular density and DWI presents restriction of water molecules, DCE is associated with increased vascularity of the tissue, which might conversely result in an improved cure rate due to the known effect of oxygenation on RT-related cell death. The accuracy of MRI-based prognostic prediction could be improved through dedicated Radiomics models. Although Radiomics are largely focused on diagnostics [38], some authors present an exceptionally high correlation of MRI features with the risk of bone metastases [39,40]. On the other hand, these results are based on rather modest study groups, prone to overfitting, and unlike PI-RADS require a measurable increase in resources necessary for each patient’s initial evaluation.

We acknowledge the limitations of the study. Although all of the patients were treated according to the same pre-specified institutional protocol, the assessment of PI-RADS score and data collection was performed retrospectively, and inclusion criteria associated with the availability of MRI and lack of pre-treatment ADT creates a significant selection bias. The use of 1.5T MRI scanners in the majority of the patients could be associated with reduced quality of PI-RADS assessment. The implementation of fusion biopsy improves the accuracy of histopathological assessment over the ultrasound-guided and in-bore MR-guided biopsy used in our patients, leading to upstaging in some cases. The low number of events limits the possibilities of statistical analysis, while a small sample size and moderate follow-up reduces the strength of the conclusions. The use of multivariable analysis was therefore limited and forced the need for synthetic minority oversampling. However, due to two-step feature selection, decision tree pruning, and complex internal validation, we believe that our models are clinically warranted and overfitting-resistant. We consider our findings to be treated as “hypothesis-generating” and encourage external validation in further research.

## 5. Conclusions

We found that PI-RADS score of 5 and size of the target lesion are associated with shorter metastasis-free survival in early-stage PCa patients and should be considered as an adverse prognostic measure for patients undergoing radiation or focal therapy. Our study suggests that patients with PI-RADS 5 and larger lesions should undergo close surveillance and/or intensification of additional diagnostics and therapy. Further prospective studies are warranted to confirm the prognostic value of the MRI features.

## Figures and Tables

**Figure 1 cancers-14-01613-f001:**
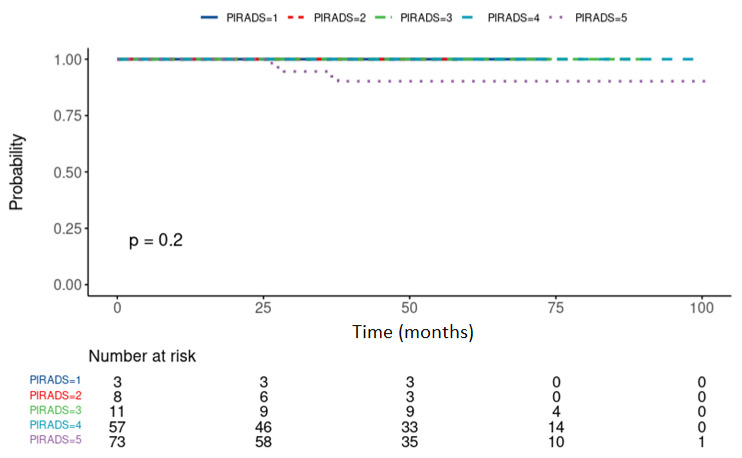
The probability of metastases-free survival over the course of follow-up depending on the PI-RADS score, in patients treated with ultra-hypofractionated CyberKnife radiosurgery for the primary treatment of a low or intermediate risk-group localized prostate cancer. All the events were experienced in patients with a PI-RADS score of 5.

**Figure 2 cancers-14-01613-f002:**
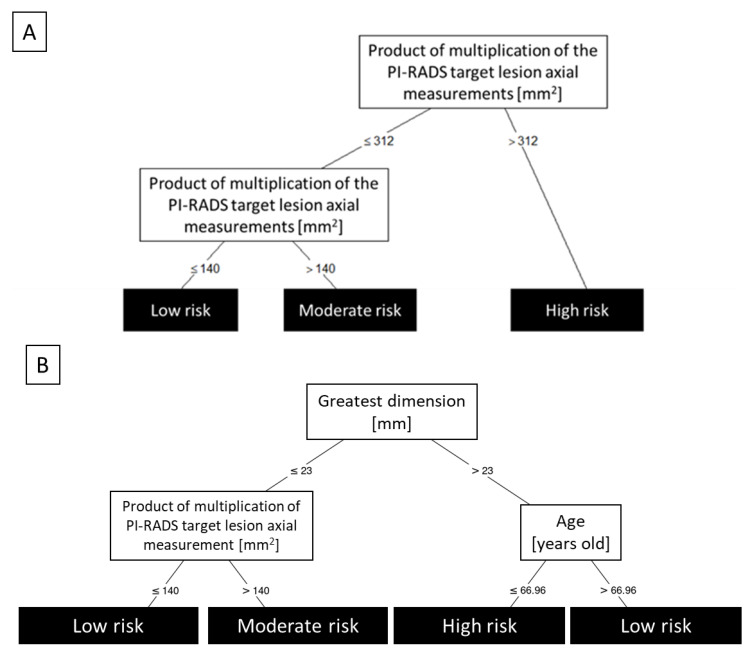
Best stratification conditional tree models. Panel **A** shows the probability of developing metastases over the course of follow-up depending on the product of multiplication of the PI-RADS target lesion axial measurements, in patients treated with ultra-hypofractionated CyberKnife radiosurgery for the primary treatment of a low or intermediate risk-group localized prostate cancer. Panel **B** provides a decision tree constructed based on the best feature subset.

**Table 1 cancers-14-01613-t001:** Clinical characteristics of the patients treated with ultra-hypofractionated CyberKnife radiosurgery for the primary treatment of a low or intermediate risk-group localized prostate cancer.

Clinical Characteristics		PI-RADS ≤ 3	PI-RADS = 4	PI-RADS = 5	Total
Number of cases		22	57	73	152
Age	(years) *	69 (64–73)	67 (62–71)	68 (61–74)	67 (62–73)
Max PSA	(ng/mL) *	7.5 (5.2–9.3)	7 (5.4–8.8)	7.7 (6.4–9.7)	7.5 (6–9.5)
Gleason Grade Group	I (3 + 3)	21 (95.5%)	52 (91.2%)	62 (84.9%)	135 (88.8%)
	II (3 + 4)	1 (4.5%)	5 (8.8%)	11 (15.1%)	17 (11.2%)
TNM T stage ^&^	T1c	14 (63.6%)	26 (45.6%)	42 (57.5%)	82 (53.9%)
	T2a	3 (13.6%)	12 (21.1%)	17 (23.3%)	32 (21.1%)
	T2b	3 (13.6%)	12 (21.1%)	10 (13.7%)	25 (16.4%)
	T2c	2 (8.6%)	7 (12.3%)	4 (5.5%)	13 (8.6%)
Risk group ^#^	Low	13 (59.1%)	28 (49.1%)	40 (54.8%)	81 (53.3%)
	Intermediate	9 (40.9%)	29 (50.9%)	33 (45.2%)	71 (46.7%)
Prostate volume	(cc) *	31.8 (26.4–35.7)	35.5 (25.2–44.4)	30.5 (25.6–40)	32.5 (25.4–40.9)
PSA density	(ng/mL/cc) *	0.23 (0.16–0.27)	0.21 (0.15–0.29)	0.25 (0.19–0.34)	0.23 (0.16–0.32)
Short-term ADT ^##^	% receiving	4 (18.2%)	5 (8.8%)	0 (0%)	9 (5.9%)
PI-RADS v2.1 components ^					
Number of lesions	1	10 (45.5%)	55 (96.5%)	66 (90.4%)	131 (86.2%)
	2		2 (3.5%)	7 (9.6%)	9 (5.9%)
Axial dimensions of the index lesion *	(1) greatest [mm] *	9.5 (7–13)	12 (10–14)	20 (17–24)	16 (12–21)
	(2) perpendicular [mm] *	5.5 (5–10)	8 (6–9)	11 (10–13)	9.5 (7–12)
	1 × 2 * [mm2]	51 (35–130)	90 (63–112)	230 (180–300)	160 (84–234.5)
Localization of the index lesion within the prostate (zone)	Peripheral	7 (31.8%)	51 (89.5%)	52 (71.2%)	110 (72.4%)
	Transitional	2 (9.1%)	5 (8.8%)	16 (21.9%)	23 (15.1%)
	Peripheral + transitional	0 (0%)	1 (1.8%)	3 (4.1%)	4 (2.6%)
	Transitional + central	1 (4.5%)	0 (0%)	2 (2.7%)	3 (2%)
Localization of the index lesion within the prostate (lobe)	Right	5 (22.7%)	22 (38.6%)	32 (43.8%)	59 (38.8%)
	Left	4 (18.2%)	28 (49.1%)	27 (37%)	59 (38.8%)
	Both	1 (4.5%)	7 (12.3%)	14 (19.2%)	22 (14.5%)

* median (IQR); ^&^ based on digital rectal examination; ^ assessment omitted for PI-RADS < 3 and one Pi-RADS = 3 cases; ^#^ D’Amico Risk Classification; ^##^ 6-months ADT, implemented post-MR.

**Table 2 cancers-14-01613-t002:** Results of best subset evaluation. The table contains 10 best subsets with the models’ c-index and internal validation (bootstrap bias-corrected) c-index with its 95% CI.

Variables Included	Training C-Index	Internal Validation C-Index	95% CI of Validation C-Index
Greatest dimension + Product of multiplication of PI-RADS target lesion axial measurements + age	0.903	0.896	0.848–0.901
Greatest dimension + Product of multiplication of PI-RADS target lesion axial measurements	0.896	0.885	0.845–0.892
Greatest dimension + T2 weighted imaging (T2W) score	0.897	0.874	0.837–0.895
Greatest dimension + age + PIRADS	0.906	0.873	0.838–0.897
Greatest dimension + PIRADS	0.900	0.870	0.845–0.900
Sum of PI-RADS target lesion axial measurements + age + PIRADS	0.913	0.863	0.797–0.877
Greatest dimension + T2 weighted imaging (T2W) score + age	0.904	0.852	0.834–0.900
Sum of PI-RADS target lesion axial measurements + age	0.902	0.847	0.804–0.878
Greatest dimension + Sum of PI-RADS target lesion axial measurements	0.893	0.843	0.842–0.891
Sum of PI-RADS target lesion axial measurements + T2 weighted imaging (T2W) score	0.874	0.839	0.820–0.865

## Data Availability

Anonymized data available on request due to privacy and ethical restrictions.

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
