# Peer review of "The Prognostic Value of PI-RADS Score in CyberKnife Ultra-Hypofractionated Radiotherapy for Localized Prostate Cancer"

_cancers, 2022, doi:10.3390/cancers14071613_

Round 1

Reviewer 1 Report

The approach of this study is of interest and important. The outcome is in agreement to what is expected from the clinical experience, which is a merit. The main weakness is, as the authors pointed out, that the modeling is based on only 5 events. A substantially longer follow-up or a larger cohort is necessary to confirm the accuracy of the prognostic modeling.

Author Response

Reviewer 1

The approach of this study is of interest and important. The outcome is in agreement to what is expected from the clinical experience, which is a merit. The main weakness is, as the authors pointed out, that the modeling is based on only 5 events. A substantially longer follow-up or a larger cohort is necessary to confirm the accuracy of the prognostic modeling.

Dear reviewer,

Thank you for your comments and positive response! We believe that introduction of Pi-Rads to the initial assessment of prostate cancer patients' prognosis can be an important step forward in personalizing treatment in the era of nearly routine initial prostate MRI, especially with patients with limited disease, and we are glad to contribute our work. The conclusions could indeed be improved through longer follow-up and a larger database. We seek to continue our work, expand the follow-up, and join efforts with other researchers around the world to create larger and more accurate models. According to the comments, we made sure to review the article once again and corrected any language errors we could find.

Reviewer 2 Report

Miszczyk and colleagues present a sound manuscript on PIRADS classification as a prognosticator in low and intermediate prostate cancer patients after stereotactic body radiotherapy, tackling the issue of insufficient risk stratification in order to tailor personalised treatment approaches. The authors included a rather small number of patients (leading to a small number of events), but present a sufficient follow-up time and show promising results, that the PIRADS score can be applied as a prognostic tool. 

The manuscript is well and precisely written and the statistical methods are accurate. In the following I highlight some aspects that need minor revision:

Introduction:

Please elaborate on the study (citation 11). Why does your study add significant value to this topic.

Methods:

Please provide more details on the used MRI scanners and protocols. Mostly 1.5 T scanners were used. PIRADS recommends to use 3T scanners, consequently additional information is necessary to evaluate the quality of the acquired images. I would suggest to mention this limtitation.

PIRADS assessment was conducted by one radiologist. How is experience defined? Where there any uncertainties? How were they resolved? Consensus with another experienced radiologist? Were local reports available or did the authors create new classifications? 

T stage was based on DRE or MRI examination?

Were the fiducially placed prior to MRI or only prior to planning CT?

MFS: In the results section the authors mention the respective diagnostic methods that detected metastases. Based on what recommendations was the method chosen? Several definitions of MFS exist. The initiative Intermediate Clinical Endpoint in Carcinoma of the Prostate define MFS as distant metastases based on conventional imaging. Consequently to better understand and compare the results presented in this study biochemical recurrence free survival rates and PSA values should be included in the mansucript and analysis.

Results:

Table 1 please add "according to NCCN" to the presented risk groups. Please emphasise that mostly favourable intermediate risk patients were included in this study.

Discussion

I would emphasise to discuss deeper MRI image analysis (for example Radiomics based analysis). How does the PIRADS score perform as a prognosticator compared to published Radiomics models?

Please elaborate on the clinical implications you suggest.

Author Response

Reviewer 2

Miszczyk and colleagues present a sound manuscript on PIRADS classification as a prognosticator in low and intermediate prostate cancer patients after stereotactic body radiotherapy, tackling the issue of insufficient risk stratification in order to tailor personalised treatment approaches. The authors included a rather small number of patients (leading to a small number of events), but present a sufficient follow-up time and show promising results, that the PIRADS score can be applied as a prognostic tool. 

Dear reviewer,

Thank you for your thorough review. We believe that introduction of Pi-Rads to the initial assessment of prostate cancer patients' prognosis can be an important step forward in personalizing treatment in the era of nearly routine initial prostate MRI, especially with patients with limited disease, and we are glad to contribute our work. We have made sure to address all your comments to our best knowledge. Please find the answers below, accompanied by respective references to changes made, which are marked in bold in the manuscript.

The manuscript is well and precisely written and the statistical methods are accurate. In the following I highlight some aspects that need minor revision:

Introduction:

Please elaborate on the study (citation 11). Why does your study add significant value to this topic.

We believe that there are two main significant differences between our studies. First of all, our study focuses on early-stage prostate cancer, while the work by Turchan et al. was mainly based on advanced-stage prostate cancer patients, many with initial lymph node involvement. Second, and foremost, our study presents a homogenous group of patients treated with a pre-defined and uniform treatment modality (only variability was 6-month concomitant and adjuvant ADT introduced in a total of 6% patients), while the work by Turchan et al. includes both patients treated with EBRT, BT, or EBRT+BT boost, patients undergoing initial AS, and finally, a variability within these groups (i.e. regarding prescribed dose, ADT and/or pelvic RT), which could significantly affect outcomes. Therefore, our findings prove that the Pi-Rads remains a prognostic factor in patients with localized prostate cancer, and after reducing the treatment-related confounding variables to a minimum. We have added such comments to the 4th paragraph of the discussion.

Methods:

Please provide more details on the used MRI scanners and protocols. Mostly 1.5 T scanners were used. PIRADS recommends to use 3T scanners, consequently additional information is necessary to evaluate the quality of the acquired images. I would suggest to mention this limtitation.

We have added this limitation in the last paragraph of the discussion section. I’ve included the names of our MR scanners in 2.2 paragraph of the Materials and Methods, along with protocols used in the Pi-Rads assessment.

PIRADS assessment was conducted by one radiologist. How is experience defined? Where there any uncertainties? How were they resolved? Consensus with another experienced radiologist? Were local reports available or did the authors create new classifications? 

The radiologist has approximately 20 years of experience, and specializes in prostate cancer imaging (including MR-guided biopsies). All the reports were created anew. Any uncertainities were resolved through consultation with another radiologist, but such were very few, and there was no dedicated second radiologist for these selected cases. I have added this information to ‘2.2. Pi-Rads score’ chapter of the Materials and Methods.

T stage was based on DRE or MRI examination?

This is a very valuable comment. The T stage was based primarily on DRE examination, I have added the information under Table 1. However, I am aware of the neverending controversy. Despite EAU guidelines, the NCCN does not explicitly state that T is DRE-only, and some of the physicians tend to grade palpable tumors as T2a-c depending on MRI findings, or even non-palpable tumors as T2 depending on medical imaging or pathology report. I’ve double-checked all T stages before analysis.

Were the fiducially placed prior to MRI or only prior to planning CT?

The fiducials were placed after MR and before CT. The treatment planning MR performed post-fiducials in remaining part of the initial study group were non-contrast and lacked dynamic series. Otherwise, it would be difficult to use anyways due to the fiducials anyways, especially in T2 series. I’ve added the information regarding MR to the methodology (2.3. Treatment protocol and follow-up).

MFS: In the results section the authors mention the respective diagnostic methods that detected metastases. Based on what recommendations was the method chosen? Several definitions of MFS exist. The initiative Intermediate Clinical Endpoint in Carcinoma of the Prostate define MFS as distant metastases based on conventional imaging. Consequently to better understand and compare the results presented in this study biochemical recurrence free survival rates and PSA values should be included in the mansucript and analysis.

This is a very important point – I have added the information that we used the occurrence of any metastases in the definition of MFS (2.3. Treatment protocol and follow-up, second paragraph). However, I’m not so sure about the biochemical endpoints. This study is challenged by two significant limitations concerning biochemical endpoints:

  • The phoenix criterion is rather liberal (nadir + 2), and relatively often that we find metastases and introduce treatment before formal biochemical recurrence with the aid of routinely available MRI and PET-PSMA. If I remember correctly – it was the case in two of the patients described in this study, which eventually did not experience formal biochemical recurrence due to introduction of salvage treatment and subsequent PSA decrease.
  • Due to the high fraction dose, some of the patients experienced PSA-bouncing, which often triggers the formal diagnosis of biochemical recurrence, but does not constitute a treatment failure.

Considering the above, I’m worried that the biochemical results might be a bit confusing. Nevertheless, the data is available. If you consider the abovementioned, and still think that it would be better to include biochemical results, we can include them it in the article (perhaps as supplementary data?).

Results:

Table 1 please add "according to NCCN" to the presented risk groups. Please emphasise that mostly favourable intermediate risk patients were included in this study.

Although I fully agree that this has to be clarified, the risk stratification system used in this study was d’Amico classification. I have added this information to the article (under Table 1 and in methods).The patients were treated in a semi-prospective manner, according to a pre-specified treatment and follow-up protocol, which included reporting of d’Amico scale risk groups. We included the PSA density as an additional factor in the analysis, but I’m not so sure about pathology-derived parts of the NCCN risk groups classification. In fact, they were not available in many patients (i.e. max % of cancer not specified, or poorly specified how many cores were positive in older pathology reports).

Discussion

I would emphasise to discuss deeper MRI image analysis (for example Radiomics based analysis). How does the PIRADS score perform as a prognosticator compared to published Radiomics models?

We have included Radiomics in the discussion, along with respective comparison and limitations. Although radiomics can be very helpful, it seems that for now there is a relative lack of standardization, and requirement for additional analysis. There are two interesting reports with regard to the probability of developing bone metastases, but based on limited material and only slightly higher c-index (0.87 vs. approx.. 0.9). The Pi-Rads excels in the aspect that it requires no further resources, as it is routinely assessed during pre-treatment MRI.

Please elaborate on the clinical implications you suggest.

We have elaborated on the clinical implications of our findings in the discussion section. We believe that PiRads assessment can be an important tool in selecting a proper patient-tailored treatment in prostate cancer patients, and allow for a better selection of proper candidates for observation.

Moreover, we have performed an additional language chceck and corrected any remaining flaws, according to the suggestion in the review questionnairie. We hope that you will find our response satisfactory, and we’re looking forward to any further comments.